# Adolescent Wilderness Therapy: The Relationship of Client Outcomes to Reasons for Referral, Motivation for Change, and Clinical Measures

**Nevin J. Harper** [1], **Will W. Dobud** [2] and **Doug Magnuson** [3,*]

1   School of Exercise Science, University of Victoria, Victoria, BC V8W 2Y2, Canada; njharper@uvic.ca
2   School of Social Work and Arts, Charles Stuart University, Wagga Wagga 2795, Australia; wdobud@csu.edu.au
3   Educational Psychology and Leadership Studies, University of Victoria, Victoria, BC V8W 2Y2, Canada
*   Correspondence: dougm@uvic.ca

**Abstract:** Outdoor behavioral healthcare is a specific model and industry utilizing wilderness therapy (WT), a residential treatment approach comprised of outdoor travel and living for youth experiencing mental health, substance use, and behavioral concerns. We present data from 6417 participants about reasons for referral, admission, and discharge scores from the Youth-Outcome Questionnaire (Y-OQ), youth interest and commitment to treatment, reliable change scores, and the relationship between these variables. One-third of youth entered WT with sub-clinical scores, varying levels of client motivation and voluntariness in relation to clinical outcomes, a diverse range of presenting problems without clear indication of specialized treatment planning, and differing responses to treatment by referral reason. Identifying those not responding to WT and those at risk of deterioration from the time of admission requires further investigation to improve client outcomes for this treatment modality. Recommendations include placing increased importance on accurate and thorough screening and assessment, utilizing baseline and routine outcome monitoring, reducing coercion, and considering specialized intervention.

**Keywords:** wilderness therapy; outdoor behavioral healthcare; psychotherapy outcome; client fit; adolescent treatment; residential youth treatment





## 1. Introduction

Youth mental health treatment is in demand, and report after report suggests a shortage of services in North America [1–3]. According to Malla et al. [4], the "current gross inadequacies in access and quality of care available" (p. 216) indicate the need for greater investment, increased prevention, and an exploration of alternatives to outpatient counseling, family support, or more intensive treatment options.

Because of the shortage, outdoor therapy from private organizations appeals to parents and clinicians when other mental health interventions are unavailable or have not elicited the desired outcome. There are numerous types of outdoor therapy for numerous types of clients, including cancer survivors [5], youth experiencing mental health concerns [6], adjudicated adolescent populations [7], those experiencing concerns of substance misuse [8], adolescent sexual offenders [9], post-traumatic stress disorder and military veterans [10], and reunifying foster care siblings [11].

In this paper, we examine one outdoor therapy option for youth—wilderness therapy (WT) [12]. We report measures of clinical effectiveness with data from outdoor behavioral healthcare (OBH), which is described below. Our aim is to provide evidence-informed clinical implications for WT practice.

### 1.1. Wilderness Therapy

Wilderness therapy consists of outdoor living and travel for multiple days (or weeks) in remote locations [13]. The treatment takes place away from settlements and structures,

in natural landscapes, and travel is via human power (i.e., by foot or paddle craft). These expeditions are usually self-contained (i.e., small groups carrying packs with food, fuel, shelter) and most often utilized for problems of adolescent substance use, mental health, and behavioral issues [13–15]. According to Russell and Philips-Miller [16], clients reported that physical exercise, outdoor living, peer feedback, group counselling, and relationships were key factors in improving their behavior during their outdoor WT experience.

Criticisms of WT include the frequent absence of outdoor settings and adventure activities as treatment variables in research models [17,18], and in the United States, the use of involuntary youth transport, coercive treatment [19,20], and a lack of family reunification [21].

Many WT research studies treat both participants and the experience as homogeneous, leading to the prescription of WT for diverse problems, including ADHD, poor school performance, and substance use disorders, without unpacking the "black box" of how it might work. For example, Harper et al. [21] found that about one-third of youth were ambivalent about their WT experience. Qualitative research with 148 youth questioned whether the wilderness experience was therapeutic [22]. Most reported the experience to be challenging, some noted the outdoors to be beneficial, and others found WT disruptive and unnecessary. A progressive research agenda, described by Wampold and Imel [23], is needed to address anomalies like these and to unpack the implications for WT practice.

### 1.2. Concerns about the Practice and Evaluation of Wilderness Therapy

Used widely in psychotherapy, the Youth-Outcome Questionnaire (Y-OQ) at admission and discharge was first described in WT research by Russell [24]. Since then, a database of Y-OQ outcomes has been managed by researchers from the Outdoor Behavioral Healthcare (OBH) Center to evaluate the effectiveness of WT programs (see obhcenter.org, accessed on 12 July 2023). The center hosted seven research scientists who contributed to more than 200 pieces of research, mostly focused on WT and adventure therapy.

In the initial studies, Russell [24] found that "at admission clients exhibited presenting symptoms similar to inpatient samples, which were on average significantly reduced at discharge" (p. 355). Magle-Haberek et al. [25] found that Y-OQ admission scores related significantly to overall change scores at discharge but were not related to the type of program a client attended or the length of the WT or other residential treatment. Gillis et al. [26] reported a growing number of WT studies with large effect sizes and argued for routine outcome monitoring so clinicians can understand the change process beyond pre/post-outcome studies. Since then, the vast majority of OBH research has reported overwhelmingly positive effects without clear details about client type, program mechanisms, staffing differences, or where youth went after graduating from the program.

For example, Gass et al. [27] described OBH outcomes as 424% better than treatment-as-usual due to OBH's 94% treatment completion rates, though they did not reveal that many participants were involuntary and not allowed to leave. DeMille et al. [28] used parent report Y-OQ data to compare OBH participants to those in a treatment-as-usual (TAU) group, finding that OBH participants were significantly better off one-year post-discharge, though they did not describe the treatment provided to the TAU group, a common critique of using treatment-as-usual for control in psychotherapy clinical trials [23].

Typically, when psychotherapy treatments are compared, they appear equally efficacious [29], and this finding has been largely ignored in the outdoor therapy literature. Dobud and Harper [17] reviewed studies comparing outdoor therapies to indoor counterparts and found small to no differences in outcomes. These large claims have not been replicated, and ongoing analysis of client, therapist, treatment, and process factors are needed to advance WT research.

Cuijpers et al. [30] found that 54% of youth receiving psychotherapy experienced clinically significant change, and 6% reported deterioration. Tucker et al. [31] reported that 50–55.8% of youth reached clinically significant change in a community-based setting utilizing adventure-based therapies, and 11.1% deteriorated. This recent work is the only study we know of in which deterioration was reported in outdoor therapies.

Rates of deterioration remain important for clinicians. Hannan et al. [32] found that clinicians were rarely able to predict who on their caseload was improving or deteriorating when outcome measures were not used to monitor client change in real-time, and this is generally true for counselors. It is essential that practitioners identify those they serve who are not benefitting and adjust the intervention or refer them on to more effective care. This is rarely performed in wilderness therapy, and in OBH programs, participants are rarely allowed to leave.

Best practices in the youth treatment literature include the need for accurate assessment, matching of individual needs and preferences with appropriate interventions, and providing safety signals for treatment deterioration or dropout throughout the process. They also include assessment of youth and only admitting those who need treatment and who are a good match for the program. That this is challenging is indicated by Garcia and Weisz [33], who found that anywhere from 28 to 75% of children and adolescents prematurely disengage from outpatient treatment, and Johnson et al. [34] found a significant relationship between dropout rates and diagnoses. Clients more likely to drop out were experiencing family problems, conduct disorder, and ADHD, and conversely, those less likely to drop out experienced anxiety disorders and adverse life events and those without a clinical diagnosis. Youth in most OBH programs are not allowed to leave, a practice that concerns us.

To summarize, differences in participants are rarely included in research models; whether participants are involuntary or not is not specified, there are claims of effectiveness that are unrealistic, and there is very little published about participants who deteriorate in treatment. There is also very little data published about what proportion of participants do well.

Our research questions are as follows:

(a) What proportion of OBH participants are admitted with Y-OQ scores above and below clinical values for total scores and for subscales?
(b) What are the diagnoses/reasons for referral of OBH participants, and what are the outcomes by reasons for referral?
(c) What proportion of OBH participants are voluntary, and how are outcomes related?
(d) What proportion of OBH participants improve and deteriorate with treatment?

Our inquiry into these research questions is informed by a pragmatic philosophy of science to explore practical outcomes that may inform clinical work in WT based on clients' self-reported outcomes and engagement; a similar approach is described in Wampold & Imel's [23] research program. We used data from OBH programs to report clinical outcomes sorted by referral reasons, youth interest in placement, commitment to participation, and intake scores. We aimed to examine trends in WT outcomes to assist clinicians in identifying who may not benefit from WT and who is deteriorating in their care.

## 2. Materials and Methods

### 2.1. Data

Data were obtained from the OBH Center in 2018, a group of WT programs and affiliated researchers whose mission is the "development of best practices, effective treatments, and evidenced-based research" (https://www.obhcenter.org/, accessed on 12 July 2023). OBH as a practice is the prescriptive use of "extended backcountry travel and wilderness living experiences long enough to allow for clinical assessment, establishment of treatment goals, and a reasonable course of treatment not to exceed the productive impact of the experience" [27].

Data acquired included demographics, survey responses from parents, youth, and staff, as well as Y-OQ treatment scores of 6417 participants attending WT programs between April 2010 and April 2018 across 16 different programs in the United States. Because of missing data, we used between 2400 and 4047 participants for each analysis presented here (i.e., we used only data with complete sets for those variables and/or questions).

### 2.2. Participants

Of 6417 participants, 4376 were male (70%), 1999 were female, and 17 were unknown. There were 4484 missing ethnicity values. Of the 1933 with data, 65% were White, 12% were

classified as "not Hispanic/Latino", and the remainder were small proportions (less than 5%) of Hispanic/Latino, African American, Native American, Pacific Islander, Asian, and Other. The mean age of participants was 15.7, with a standard deviation of 1.2 and a range of 13–18. Family incomes were typically over USD 100,000 per household per year. On average, WT in the USA costs USD 558 per day as well as a one-time enrollment fee of approximately USD 3194 (https://www.allkindsoftherapy.com/, accessed on 17 October 2023), making the cost of a stay, in this sample, a mean of 89 days, about USD 52,856 plus transportation and educational consultant fees if those services were utilized. About half of the participants in this sample were transported by third-party service providers to OBH programs.

*2.3. Measures*

2.3.1. Youth-Outcome Questionnaire (Y-OQ) 2.01 (SR)

The Y-OQ is a 64-item self-report measure designed to identify presenting problems across six domains: Interpersonal Distress, Somatic, Interpersonal Relations, Social Problems, Behavioral Dysfunction, and Critical Items [35]. It is a self-report form for 12–18-year-olds. The Y-OQ has been shown to be an effective clinical tool for monitoring change trajectories of psychological disorders across multiple domains of the client's life, including identifying progress, providing safety signals for drop-out or deterioration, and effecting increased change processes [36]. The Y-OQ total scores under 47 are considered non-clinical (i.e., community normative range), and 47 and above are clinical (240 maximum scores) with a Reliable Change Index (RCI) of 13. That is, changes in scores on the RCI of 13 or more suggest a clinically relevant improvement or worsening in client functioning (i.e., reliable change).

2.3.2. Youth-Outcome Questionnaire Subscales

*Intrapersonal Distress (ID):* "Anxiety, depression, fearfulness, hopelessness, and self-harm" [37] are included here. Higher scores indicate increased emotional distress. The clinical cutoff is 17.

*Somatic (S)*: Physical symptoms include "headaches, dizziness, stomach aches, nausea, bowel difficulties, and pain or weakness in joints" [37]. Higher scores indicate increased distress. The clinical cutoff is 6.

*Interpersonal Relations (IR):* These questions focus on relationships with parents, adults, and peers, and higher scores indicate increased difficulties in managing them. The clinical cutoff is 3.

*Social Problems (SP)*: "This scale assesses troublesome social behaviors" [37]. The scale is focused on violations of social norms, for example, running away from home or substance abuse. Higher scores indicate increased difficulties. The clinical cutoff is 3.

*Behavioral Dysfunction (BD)*: "This scale assesses change in the child/adolescent's ability to organize tasks, complete assignments, concentrate, and handle frustration, including times of inattention, hyperactivity, and impulsivity" [37]. Higher scores indicate increased dysfunction. The clinical cutoff is 11.

*Critical Items (CI):* This scale "addresses paranoia, obsessive-compulsive behaviors, hallucination, delusions, suicide, mania, and eating disorders" [37]. Higher scores indicate increased problems and a high score for single items suggests a serious concern. The clinical cutoff is 6.

2.3.3. Questionnaire Items

*Reason for Referral.* Staff completed a questionnaire at participant admission and wrote in answers to the question, "What is the client's primary reason for referral?" There were two additional questions to identify the secondary and tertiary reasons for admission. We used the primary and secondary reasons. Staff selected from a list of closed-ended options that included Alcohol/substance abuse, Anxiety, Attention issues (ADHD/PTSD), Autism/Asperger's, Depression, Mood Disorder, Learning Disability, Oppositional Defiant/Conduct, Trauma-related, Eating disorders, Self-injury, Obsessive-compulsive disor-

ders, or Other. If Other was selected, a word or phrase was inserted. Whether these are diagnoses passed on by experts or labels given by parents to staff is not known.

*"It makes sense for me to be in a therapeutic program"*. This question was asked of participants at admission on a scale from 0 to 100. The scale was represented as a continuous line. Likert item labels were written above the line, with 75 and above corresponding to Agree and Strongly Agree and below 75 corresponding to Neutral, Disagree, and Strongly Disagree. For this paper, we consider this question and the following question as indicators of client engagement in the therapeutic process at the beginning of their care, albeit imperfect.

*"How do you feel about being at this program?"* Participants were asked this question at admission, and the choices were 1: very negative, 2: negative, 3: neutral, 4: positive, and 5: very positive.

*"Was your child transported?"* Parents responded to this question, yes or no. Transport usually entails involuntary delivery of the youth by a third-party provider who picks up youth, often from their homes at night, and brings them to WT programs [20]. This question was criticized for its use by researchers as a proxy for *voluntary* [21]. It is a single-item question, and it does not directly ask whether participants were, in fact, voluntary. It is likely there are more involuntary participants than indicated by this question, that is, youth who resisted or did not agree with the treatment choice but were transported by their parents. Despite these concerns, we used it here to illustrate the distribution of participants as a rough substitute for voluntary status.

*2.4. Data Analysis*

For each research question, participants with missing values were deleted. No imputation was conducted. For example, for one of the Y-OQ comparisons, only the admission and discharge scores were selected, and only participants whose data were complete for both admission and discharge were included. As a result, the frequencies vary between each analysis.

First, to show the diversity of treatment issues, we tallied the number of participants by primary and secondary referral reasons. Second, to show the wide range of participant interest in engaging with programs, we constructed a scatterplot of participant responses to the two related questions about how they feel about being at *this* therapeutic program and whether it makes sense to be in *a* therapeutic program, and we compared those transported to those who were not. Third, we examined the number of participants presenting above and below the clinical cutoff of the Y-OQ total score at admission. Fourth, we utilized the "reliable change index" (RCI) scores for all participants (Discharge minus Admission) on the Y-OQ to sort them into three categories: reliable improvement, no reliable change, and reliable worsening. We tallied the numbers in each category by admission scores above and below the clinical cutoff values for the Y-OQ total score. Fifth, we computed the proportions of participants who improved, showed no change, or deteriorated, sorted by primary referral reason. Finally, we examined the RCI differences between participants' answers to the question "It makes sense to be in a therapeutic program" and their admission Y-OQ total score.

**3. Results**

Table 1 shows that the Y-OQ total score and subscale mean values for all participants are above the clinical cutoff at admission and below at discharge. The robust pre-post scores are positive.

However, there were between 1239 (31.6%) and 1814 (46.2%) participants above the clinical value at discharge across the categories. These are high percentages and high frequencies, and these suggest realistic outcomes more often seen in other youth treatment literature. Critical Items, for example, with a total discharge mean of 5.3, below the critical value, also had 1547 youth with scores above the cut-off (39.4%). The Y-OQ manual suggests that clinicians use Critical Items (suicidal ideation, hallucinations, paranoia, delusions) to monitor participant well-being because of the serious nature of these items and to intervene

where necessary. Higher scores require more immediate clinical attention, yet none of the literature is about the use of this measure in OBH programs. Significant missing values at admission and discharge raise questions about client monitoring and client-care decisions.

**Table 1.** Y-OQ clinical values, admission median, discharge median, and clients above clinical values at discharge across subscales.

| n = 6417 | Clinical Values | Admission Mean | Discharge Mean | Clients Above Clinical |
| | | Missing Values = 1593 | Missing Values = 2491 | Value at Discharge (n) |
| --- | --- | --- | --- | --- |
| Y-OQ Total score | $\geq$47 | 67 | 40 | 1458 |
| Interpersonal Distress | $\geq$17 | 23.6 | 14 | 1384 |
| Somatic | $\geq$6 | 7.5 | 5 | 1239 |
| Interpersonal Relations | $\geq$3 | 5.6 | 1.9 | 1330 |
| Social Problems | $\geq$3 | 7.7 | 3.5 | 1750 |
| Behavioral Dysfunction | $\geq$11 | 14.5 | 10 | 1840 |
| Critical Items | $\geq$6 | 8.2 | 5.3 | 1547 |

### 3.1. Reasons for Referral

Table 2 shows the primary and secondary reasons for referral identified by staff. Depression/Mood Disorders (n = 1172), Alcohol/Substance Abuse (n = 876), and Oppositional Defiant (n = 704) were the three most common primary reasons identified, and Anxiety (n = 825) was the highest secondary reason for referral.

**Table 2.** Reasons for referral.

| Variable | Primary Reason (n) | Secondary Reason (n) |
| --- | --- | --- |
| Alcohol/Substance Abuse | 876 | 674 |
| Anxiety | 529 | 825 |
| Attention Issue (ADD/ADHD) | 376 | 368 |
| Autism/Asperger's | 140 | 49 |
| Depression/Mood Disorder | 1172 | 924 |
| Learning Disability | 29 | 64 |
| Oppositional Defiant | 704 | 659 |
| Other | 488 | 663 |
| Trauma-Related Issue (PTSD) | 284 | 144 |

Reasons for referral in the *Other* category include anger, attachment, learning troubles, family troubles, issues of personal accountability, self-esteem/identity concerns, suicidal ideation, and some that were the same as the named closed-end categories, such as trauma, substance use, and ADHD. Most of these categories were not clearly defined by OBH programs, and most do not have a clear connection to a mechanism of change or ideal approach in WT.

### 3.2. Client Motivation and Willingness at Admission

Participants were asked at admission to rate their agreement with the question, "It makes sense for me to be in a therapeutic program", and 75 and above believed it made sense, while those below 75 were neutral or disagreed. This is the x-axis. The y-axis is the answers to the question, "How do you feel about being at this program?" The first question asks about whether they agree they should be in *a* program, and the second asks about being in *this program* in particular.

Figure 1 is a scatterplot of these two variables plotted on the x- and y-axes. The data are displayed by whether participants were transported—yes or no. There is a trend for transported youth to score lower on both measures.

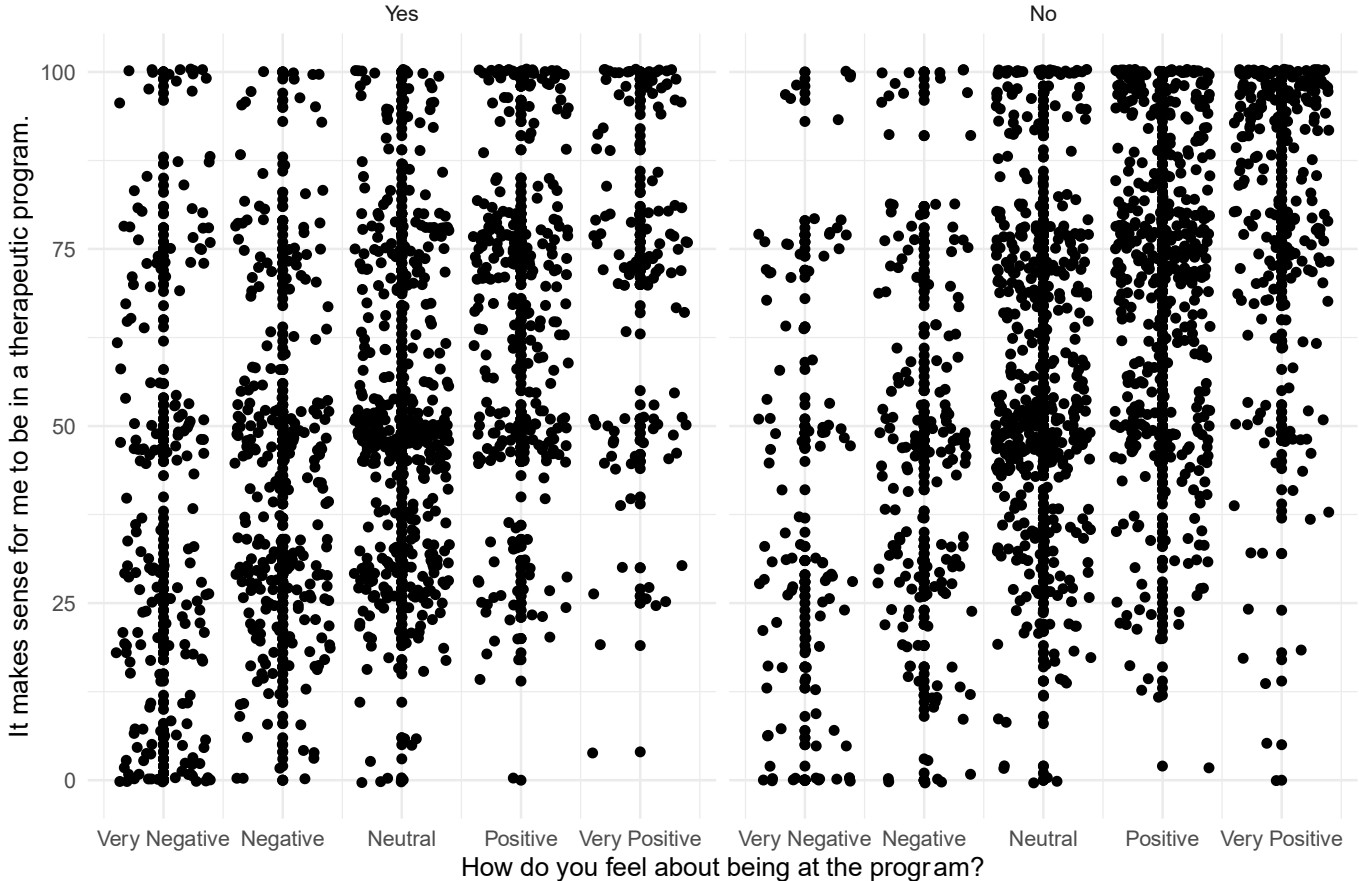

**Figure 1.** How clients feel about being in therapy and at the program.

Motivation and Willingness

1. There were 2550 who scored roughly less than "agree" on "It makes sense for me to be in a therapeutic program" at admission. There was considerable uncertainty among participants about whether they should be in *a* therapeutic program.

2. A total of 2599 participants scored 3 or below in response to the question, How do you feel about being at this program? They were also not sure at admission about their presence at *this* particular program.

3. There were 1790 who scored less than "agree" on "It makes sense for me to be in a therapeutic program" AND Neutral or less on "How do you feel about being at the program".

4. A total of 478 scored Negative or Very Negative on "How do you feel about being at the program" and under 40 on "It makes sense for me to be in a therapeutic program".

These findings suggest a large number of participants are at least ambivalent about admission and treatment. In the next section, we further examine those participants who (a) at admission did not score high enough to justify admission and (b) whose participation in treatment was not successful, as measured by the Y-OQ.

### 3.3. Y-OQ Total Score and Subscale Scores

Table 3 shows the number of participants whose admission scores were below the clinical value on each Y-OQ subscale and/or the Y-OQ total score. Here, we find nearly 29% of 4824 clients below the clinical cut-off value on the Y-OQ total score and 20–40% of clients

below the clinical cut-off value across the subscale items at admission. Clinically, client Y-OQ total scores between -16 and 46 are referred to as being in the "Community Normative Score Range" [37], but for ease of reading, we refer to them herein as *non-clinical* or *below clinical cutoff*. In this sample, there are 1387 youth entering WT in the non-clinical range.

**Table 3.** Number of participants with Y-OQ scores below the clinical value at admission.

| Y-OQ Subscale * | Number Scoring Less than Clinical Value (n = 4824) | Percentage below the Clinical Value |
|---|---|---|
| Interpersonal Distress | 1521 | 32 |
| Somatic | 1932 | 40 |
| Interpersonal Relations | 1481 | 31 |
| Social Problems | 982 | 20 |
| Behavioral Dysfunction | 1451 | 30 |
| Critical Items | 1791 | 37 |
| Total Score | 1387 | 29 |

* The frequencies are duplicated: a participant may have a score in more than one row.

Next, we compared participant admission Y-OQ total score to the Reliable Change Index (RCI, 13 or more) and the Y-OQ total score at discharge (see Table 4).

**Table 4.** Admission Y-OQ total scores by outcome: reliable improvement, no reliable change, or reliable worsening as per Reliable Change Index.

| | Outcome Condition * | n | Percentage |
|---|---|---|---|
| **Participants with Scores 47 and Above at Admission (n = 2434)** | Reliable Improvement | 190 | 78 |
| | No Reliable Change | 367 | 15 |
| | Reliable Worsening | 167 | 07 |
| **Participants with Scores Below 47 at Admission (n = 958)** | Reliable Improvement | 378 | 39 |
| | No Reliable Change | 381 | 40 |
| | Reliable Worsening | 199 | 21 |

* Outcome by RCI based on pre-to-post Y-OQ total score difference.

If one's Y-OQ total score is at or above the clinical score (i.e., 47) at admission, the percentage of clients improving is 78. If one's Y-OQ total score is below the clinical cutoff at admission, one's probability of improving is about 39%. Of significant clinical concern is finding the rates of clients entering WT with below clinical scores who experienced no change (40%) or deteriorated (21%).

The sample size is smaller in this analysis compared to Table 3 (n = 4824) due to fewer participants completing both admission AND discharge Y-OQ measures. We will discuss this further below as clients not completing Y-OQ at admission and discharge have not then completed a clinically monitored intervention as per the measure's design, that is, not having at least two "measures" to compare. As described in the literature, clinicians may not know if clients improved, deteriorated, experienced no change, or whether significant issues are present on Critical Items, despite Critical Items requiring the highest priority in treatment planning [37]. Further, regardless of outcome or assessment, youth are then leaving WT, possibly returning to their home and community, or more likely transitioning into further residential care [21].

### 3.4. Outcomes by Referral Reason for All Participants

We compared RCI scores by primary referral reason using the closed-ended question categories. Table 5 shows the proportions of participants (i.e., by referral reason) who

demonstrated Reliable Improvement, No Reliable Change, and Reliable Worsening. The frequencies for each category do not add up to the same numbers as in Table 2 above because of missing discharge scores for some participants. We find the highest proportions of improvement for Alcohol/Substance Use (0.7), Anxiety (0.71), and Depression/Mood Disorders (0.71). The highest proportions of deterioration were found for Autism/Asperger's (0.29), Attention Issues (ADD/PTSD) (0.15), and Trauma-Related Issues (0.14). From a clinical perspective, these findings suggest potential contraindications for WT or a need for specialized treatment. Our findings for clients with Trauma-Related Issues, for example, show a near 50/50 likelihood of treatment success versus no change and deterioration.

**Table 5.** Proportion of participants at discharge who demonstrate reliable improvement, no reliable change, and reliable worsening by primary reason for referral.

| Reason | Reliable Improvement | No Reliable Change | Reliable Worsening |
| --- | --- | --- | --- |
| Alcohol/Substance Use (n = 620) | 0.70 | 0.21 | 0.09 |
| Anxiety (n = 328) | 0.71 | 0.20 | 0.09 |
| Attention Issue (ADD/PTSD) (n = 239) | 0.63 | 0.22 | 0.15 |
| Autism/Asperger's (n = 66) | 0.47 | 0.24 | 0.29 |
| Depression/Mood Disorder (n = 755) | 0.71 | 0.19 | 0.10 |
| Oppositional Defiance (n = 442) | 0.64 | 0.24 | 0.12 |
| Trauma-Related Issue (n = 138) | 0.54 | 0.33 | 0.14 |
| Other (n = 372) | 0.69 | 0.22 | 0.09 |

*3.5. Admission Y-OQ Total Score and "It Makes Sense to Be in a Therapeutic Program" by RCI Outcome*

It makes intuitive sense that if a participant agrees to needing or wanting treatment, they respond better, and it also makes sense that participants who score higher at admission on the Y-OQ (i.e., higher distress) are more likely to improve. Table 6 shows that the median scores on "It makes sense. . ." who also have higher Y-OQ admission scores are those who improved. The median scores for those who experienced no change or deterioration were below the Y-OQ clinical cut-off value. It is possible the large differences in admission Y-OQ total score are, in part, a result of the probability that higher levels of distress lead towards more improvement or change. It is also possible that clients with low Y-OQ total scores at admission (i.e., below or near clinical cut-off) are not experiencing positive change, not finding meaning in treatment, or deteriorating.

**Table 6.** RCI outcome by "It Makes Sense to be in a Therapeutic Program" and admission Y-OQ.

| n = 2664<br>RCI Outcome | It Makes Sense to be in<br>a Therapeutic Program (Median) | Admission Y-OQ<br>(Median Score) |
| --- | --- | --- |
| Reliable Improvement | 59 | 75 |
| No Reliable Change | 50 | 45 |
| Reliable Worsening | 50 | 45 |

In our discussion to follow, we consider how these intake measures can inform clinical WT practice and reduce rates of deterioration and dropout, hence improving WT client service and outcomes.

## 4. Discussion

OBH researchers have made broad claims for the success of these WT programs, claims that portray them as far superior to other youth treatment interventions. As described above, Gass et al. [27] said that OBH delivers "424% better treatment outcomes as measured by the Youth-Outcome Questionnaire (Y-OQ) research instrument" (p. 1). Tucker

et al. [38] found "the gains in the OBH treatment group were significantly greater than the TAU [treatment as usual] comparison group, almost three times larger in fact" (p. 245). While there are methodological concerns when comparing specific treatments like WT to ambiguous comparison groups, such as TAU [17,23], the data from the present study suggests that only more modest claims are supported. For example, roughly 30% to 35% of all participants were discharged with Y-OQ Total Scores and/or subscale scores above the clinical cut-off values.

Second, the Critical Items subscale was intended to be used to alert practitioners to individual mental health problems that require critical attention, and there is no evidence here that these were acted upon. Third, the reasons for referral vary widely, and the literature does not provide evidence for WT as the preferred treatment for some of these referral reasons. Further, the Y-OQ subscale items, which suggest serious problems when participants score above critical values, are not theoretically or practically connected to the reasons for referral. We found that 20 to 40% of participants scored below the clinical cut-off values for the Y-OQ Total Score and/or the subscale items at admission. It appears that Y-OQ scores are used for measuring outcomes but not informing the appropriateness of care. If treatment decision-making were informed by the admission outcome data, these youth might have been treated in a possibly less intensive or less invasive modality such as outpatient counseling in their home community. In addition, only 39% of participants whose admission scores were below the cut-off values had meaningful changes at discharge, according to the RCI.

Fourth, more positively, four-fifths of participants whose admission Y-OQ scores were above the clinical cut-off reported a significant RCI. There is a meaningful difference in the experience of those with high admission scores and those with low admission scores, common to outcomes found across youth residential treatment literature.

Fifth, roughly half of these participants were transported to the program involuntarily, and the ethics of this phenomenon were addressed elsewhere [20]. Associated with this, as many as half of the participants at admission were not sure they should be admitted to a therapeutic program, or specifically this therapeutic program and large proportions of these were quite certain they should not be there. This is an ethical and practical problem. Inversely, those whose admission Y-OQ scores were high were more likely to report that it made sense for them to be admitted and were more likely to report significant change scores. Participants' sense of their own distress seems to be a good indicator of their treatment outcomes.

Finally, about 70% of participants referred for substance use, anxiety, or depressive symptoms had significant change scores. This is a realistic claim of success when compared to the meta-analytic outcome data, though we are not sure that the expense or invasiveness of this approach is always justified.

Most youth admitted to OBH are reported to have experienced previous treatment failures [20]. As we report here, a substantial proportion do not improve or get worse even though dropping out appears rare, as seen with OBH's 94% treatment completion rates [27].

When indicators of no change and deterioration are present, and the intervention is likely to have no impact or make things worse, termination of treatment should be considered. The Maine Clinician Manual [37] suggests practitioners undergo a significant treatment plan review when a client reports deterioration and obtain an objective understanding of the client's perception of progress. Routine outcome monitoring affords the client, parents, clinicians, and program staff in WT increased opportunities to better determine client fit and improve efficacy and ethical practice [39]. A progressive research program tied with data-driven decision-making, as described by Wampold and Imel [23], should focus on improving outcomes and client experiences, reducing deterioration, and, considering the benchmarks provided in our analysis, considering which youth are best suited based on outcome and preferences for WT.

## 5. Limitations

Like previous WT studies, our examination of client presentation and outcome variables includes a limited discussion of the mechanisms or pathways to change relevant to practice variables, which in themselves can be quite diverse. Further, these data are from a sample of mostly White, upper-middle-class participants in the US, and our interpretation is a summary of descriptive statistics which could inform but not predict future performance. The interpretation of our findings is also limited by a potential maturation effect occurring over the average 89 days of treatment, the healthy controlled environment provided to youth (i.e., nutritious meals, physical exercise, and a toxin/substance-free environment), and the totalistic nature of the intervention in which youth are unlikely able to leave (i.e., coercion, fear, mistrust). Future research should consider how to make use of outcome data and available evidence prior to admission (i.e., to establish a baseline before admission) and consider how client presentation impacts service delivery. Practitioners are encouraged to use routine outcome monitoring to identify at-risk cases and reduce deterioration rates such as those presented in this study.

**Author Contributions:** Conceptualization, N.J.H., W.W.D. and D.M.; methodology, N.J.H., W.W.D. and D.M.; software, N.J.H., W.W.D. and D.M.; validation, N.J.H., W.W.D. and D.M.; formal analysis, N.J.H., W.W.D. and D.M.; investigation, N.J.H., W.W.D. and D.M.; resources, N.J.H., W.W.D. and D.M.; data curation, N.J.H., W.W.D. and D.M.; writing—original draft preparation, N.J.H., W.W.D. and D.M.; writing—review and editing, N.J.H., W.W.D. and D.M.; visualization, N.J.H., W.W.D. and D.M.; supervision, N.J.H., W.W.D. and D.M.; project administration, N.J.H., W.W.D. and D.M.; funding acquisition, N.J.H., W.W.D. and D.M. All authors have read and agreed to the published version of the manuscript.

**Funding:** This research received no external funding.

**Institutional Review Board Statement:** The study was conducted in accordance with the Declaration of Helsinki and approved by the Institutional Review Board at the University of Victoria (protocol #18-169, 03.05.2018) for studies involving humans.

**Informed Consent Statement:** Archival and anonymized data was accessed through the Outdoor Behavioral Healthcare Research Centre and provided to the research team with no identifying details of clients.

**Data Availability Statement:** The authors do not have permission to share data. Data inquiries should be directed to Outdoor Behavioral Healthcare Research Center.

**Conflicts of Interest:** The authors declare no conflicts of interest.

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
