# Peer review of "Adolescent Wilderness Therapy: The Relationship of Client Outcomes to Reasons for Referral, Motivation for Change, and Clinical Measures"

_2673-995X, doi:10.3390/youth4010027_

Round 1
Reviewer 1 Report
Comments and Suggestions for Authors
The paper provides empirical evidence regarding the effectiveness of Adolescent Wilderness Therapy for addressing mental health and related issues in adolescents. The use of the Youth Outcome Questionnaire (YOQ) from over 6,000 participants from 2010 to 2018 increases the robustness of the analysis and conclusions of the study.
The findings are clear and compelling. The study reports the circumstances under which Adolescent Wilderness Therapy is most effective. Notably, participant attitudes significantly impact outcomes. When adolescents willingly acknowledge their need for or desire for treatment, positive results are more likely. In addition, Adolescent Wilderness Therapy demonstrates efficacy in specific areas, including Alcohol/Substance Use, Anxiety, and Depression/Mood Disorders. However, for certain issues such as Autism/Asperger’s, Attention Issues (ADD/PTSD), and Trauma-Related Issues, more specialized approaches may be preferable.
The data analysis section is both clear and compelling. The charts and graphics are clear and facilitate the understanding of the key findings. The discussion section is comprehensive and critically evaluates the Adolescent Wilderness Theory. It highlights limitations, objective criteria for program suitability (e.g., when participants find it personally meaningful), and ethical considerations (e.g., involuntary program enrollment).
The limitations section acknowledges that while the research provides valuable information, it does not predict future performance. Furthermore, it emphasizes the ongoing need for exploration and refinement in this field.
In summary, this paper contributes valuable empirical evidence for parents and medical practitioners, informing them about the benefits, limitations, and areas of concern involved with Adolescent Wilderness Therapy for addressing mental health and related challenges faced by adolescents.
Author Response
Thank you for the kind words.
We have made a few changes in response to the reviews, and a list with our explanation is attached. here.

Reviewer 2 Report
Comments and Suggestions for Authors
Please see attached letter.

Author Response
Thank you for your excellent suggestions.
Attached you should find a document with an explanation of our edits. We have accepted all of your suggestions, with the exception of one: the request for more theory. We are challenging some common beliefs about a particular type of wilderness therapy, and we are not sure it needs more theory. But we have added a brief statement just before the Methods.

Round 2
Reviewer 2 Report
Comments and Suggestions for Authors
To the Authors:
Thank you for your responses and edits regarding our concerns. We were very pleased to see the improvements to your paper and support publication at this time.